# Current Biomaterial-Based Bone Tissue Engineering and Translational Medicine

**DOI:** 10.3390/ijms221910233

**Published:** 2021-09-23

**Authors:** Jingqi Qi, Tianqi Yu, Bangyan Hu, Hongwei Wu, Hongwei Ouyang

**Affiliations:** 1Dr. Li Dak Sum & Yip Yio Chin Center for Stem Cells and Regenerative Medicine, Department of Orthopedic Surgery of the Second Affiliated Hospital, Zhejiang University School of Medicine, Hangzhou 310003, China; jingqi.19@intl.zju.edu.cn; 2Zhejiang University-University of Edinburgh Institute, Key Laboratory of Tissue Engineering and Regenerative Medicine of Zhejiang Province, Zhejiang University School of Medicine, Hangzhou 310003, China; 3Department of Mechanical Engineering, Zhejiang University-University of Illinois at Urbana-Champaign Institute, Zhejiang University, Haining 314400, China; Tianqi.19@intl.zju.edu.cn; 4Section of Molecular and Cell Biology, Division of Biological Sciences, University of California San Diego, La Jolla, CA 92093, USA; bahu@ucsd.edu; 5Department of Sports Medicine, Zhejiang University School of Medicine, Hangzhou 310003, China; 6China Orthopedic Regenerative Medicine Group (CORMed), Hangzhou 310003, China

**Keywords:** stem cell, scaffold, growth factor, osteogenesis, angiogenesis

## Abstract

Bone defects cause significant socio-economic costs worldwide, while the clinical “gold standard” of bone repair, the autologous bone graft, has limitations including limited graft supply, secondary injury, chronic pain and infection. Therefore, to reduce surgical complexity and speed up bone healing, innovative therapies are needed. Bone tissue engineering (BTE), a new cross-disciplinary science arisen in the 21st century, creates artificial environments specially constructed to facilitate bone regeneration and growth. By combining stem cells, scaffolds and growth factors, BTE fabricates biological substitutes to restore the functions of injured bone. Although BTE has made many valuable achievements, there remain some unsolved challenges. In this review, the latest research and application of stem cells, scaffolds, and growth factors in BTE are summarized with the aim of providing references for the clinical application of BTE.

## 1. Introduction

Bone diseases and their complications, which account for half of the chronic diseases in people over 50 years old [1], still face significant clinical challenges. With an incidence of approximately 15 million fracture cases per year worldwide, the repair and regeneration of bone has attracted extensive research in bone tissue engineering (BTE) [2]. BTE is a frontier cross-disciplinary subject in the field of life science in the 21st century composed of bioengineering, cell transplantation, and material science with the aim of constructing biological substitutes for the restoration of injured bone [3]. 

Nowadays, treatments of bone defects mainly include synthetic bone void filler, allografts, autografts, distraction osteogenesis, insertion of the vascular bundle and cement casting. Among therapeutic strategies, autografting is considered the “gold standard”; it involves harvesting the bone from one side of the patient and transplanting it into the injured area of the same patient for bone repairment [4]. However, the autograft method exhibits some limitations, such as limited graft supply, chronic pain, high donor site morbidity, secondary damage and infections [5], leading to unsatisfactory surgical outcomes. Allografts represent about 34% of bone substitutes, and their bone supplies are from donors, which are available in various sizes and are free from donor site morbidity compared with autografts [6]. However, allografts are limited by immunological rejection and infectious disease transmission, and its demand already far exceeds available supplies. In addition to biological grafts, bioinert materials, such as alumina, stainless steel, and poly-methyl methacrylate, have also been used in bone surgery [6]. Regardless of their availability and reproducibility, these materials cannot fuse well with the host bone and may be wrapped in fibrous tissue after implantation. Moreover, the stiffness mismatch between the weight-bearing implant and adjacent bone also restricts the use of bioinert grafts [6]. Altogether, the limitations of current clinical treatments appeal to novel methods for reducing surgical complexity and accelerating bone regeneration. 

By combining stem cells, scaffold, and growth factors (GFs), BTE builds optimal biomimetic environments to promote the regeneration and growth of normal tissues and cells (Figure 1). In this way, BTE can not only repair the damaged bone, but also overcome the limitations of current clinical treatments. In BTE, vascularization is one of the biggest challenges because the distance of nutrient and waste exchange is limited to 100~300 μm between individual cells and capillaries in bone [7]. Many studies have focused on the design of grafts around this diffusion limit. A promising strategy combined a cell-loaded hydrogel and a synthetic engineered vascular graft in a layered manner into a porous rigid framework of bone conduction [8]. Thus, coordinating the properties of different components to implant a perfusable vessel into a designed bone structure is promising. Following up, due to the unique physical properties of bone, a proper combination of materials with different mechanical properties needs to be considered. Presently, numerous achievements of BTE in tissue regeneration and engineering have been made, and some of the products of BTE have been applied in clinical treatments of endodontics, craniomaxillofacial applications, and periodontal regenerative therapy [9]. 

Briefly, stem cells can promote osteogenesis and angiogenesis with the stimulation of GFs, and biocompatible scaffolds can mimic the ecology of bone extracellular matrix (Figure 1). In this review, we demonstrate the latest research and clinical applications of BTE in the aspects of stem cells, scaffolds, and GFs to offer insights into BTE.

## 2. Stem Cells

Regarded as the cornerstone of BTE, stem cells are capable of self-renewing and differentiate into at least one type of offspring. In BTE, ideal candidates should meet the following criteria: (1) abundant sources and convenient sampling; (2) strong ability of in vitro cell passage with an immobile phenotype; (3) high adaptability to the environment of the receiving zone; (4) capacity to replace the missing cells and restore the tissue function; (5) safe clinical application [10]. Commonly applied stems cells in BTE are mesenchymal stem cells (MSCs), endothelial progenitor cells (EPCs), and induced pluripotent stem cells (iPSCs).

### 2.1. Mesenchymal Stem Cells (MSCs)

MSCs have been considered as the ideal seed cells of BTE for their high availability, rapid proliferation, and special functions [11]. MSCs are heterogeneous and distributed in various tissues such as muscle, adipose tissue, and bone marrow (Figure 2A). MSCs participate in various human physiological processes, including immunomodulatory processes, anti-apoptosis processes, and angiogenesis (Figure 2B). MSCs are significant for bone healing, especially in nonunion fractures caused by trauma, inadequate blood supply, or other conditions. Currently, MSCs have been used in clinical treatments of oral and maxillofacial defects, as well as long bone defects [12]. Conventionally, bone marrow-derived mesenchymal stem cells (BMSCs) are the “gold-standard” cell source for BTE clinical application. The pluripotency, anti-inflammation, immunomodulation, and hematogenic and angiogenic promotion of BMSCs facilitate their clinical involvement in the repair of long bone, vertebrae, and craniofacial bone [13]. However, the application of BMSCs has been impeded by the invasive harvesting procedure.

Adipose-derived stem cells (ASCs), another promising candidate in BTE, have been considered for their similar osteogenic capacity as BMSCs, simplicity to harvest surgically, and abundant sources. Compared with BMSCs, adipose tissue is more affluent under local anaesthesia, and lipotomy is less invasive than bone marrow aspiration, mitigating patient pain [14]. While 6 × 103 BMSCs per ml of bone marrow can be extracted [15], 2 × 105 ASCs per gram of adipose tissue can be isolated, in contrast [16]. Moreover, ASCs can maintain phenotypes for a longer time during culture with strong proliferative ability, so they are more suitable for allotransplantation than BMSCs [17]. Clinically, a case report of a 7-year-old child using autologous ASCs to repair post-traumatic skull defects showed a nearly completely continuous skull formed 3 months after surgery [18]. Currently, ASC-based therapy for osteogenesis has been reported to achieve positive results for craniofacial bone defects [19] and long bone defects [20], and the intra-articular injection of ASCs is regarded as a safe therapeutic alternative for severe knee osteoarthritis patients [21].

Genetic engineering has attracted interest to further improve the efficacy of MSCs in BTE. According to Oryan et al. [22], osteogenic genes can be transduced into MSCs through a gene-activated matrix (GAM), which is composed of a collagen scaffold impregnated with plasmid DNA-encoding osteogenic genes (Figure 2A). GAM can be inserted into bone defects, and, subsequently, the host MSCs entering the scaffold will be transfected by plasmids and produce the osteogenic gene product. Umebayashi et al. suggested that GAM with atelocollagen containing cDNA of BMP-4 or Runx2 possesses a dose-dependent osteoinduction potential on cranial bone defects in rats [23]. Additionally, as BMP-2 promotes the differentiation of osteoclasts, it has been applied to modify MSCs in the treatment of fracture and femoral head necrosis [22,24]. An in vivo study suggested that the implantation of BMSCs infected with BMP-2 could form orthotopic bone in mouse hindlimbs and repair critical-sized radial defects in a mouse [25]. Moreover, Peng et al. found that genetically modified MSCs are able to produce both osteogenic and angiogenic GFs [26]. Nevertheless, MSC gene therapy has the limitations of safety and regulatory hurdles.

### 2.2. Endothelial Progenitor Cells (EPCs)

Angiogenesis is essential for BTE, as bone is highly vascularized, relying on tight spatial and temporal connections between blood vessels and osteocytes to maintain integrity. Adequate vascularization is a prerequisite for stem cells to reach the site of the defective tissue and to obtain oxygen and nutrients (Figure 1) [27]. EPCs have the ability to differentiate into endothelial cells and participate in angiogenesis [28]. The effectiveness of EPCs in inducing angiogenesis has been demonstrated in animal studies of cardiovascular disease, peripheral vascular disease, and stroke [29]. Moreover, the promotion of EPCs in bone repair and regeneration through neovascularization has been demonstrated in preclinical trials. EPCs have been found to secrete osteogenic factors such as BMP-1, 2, 3, 6, 7, 8 and TGF-βs that significantly enhance MSC-induced osteogenesis [30]. Through autologous EPC implantation, dense woven bone was formed in a critical-sized tibia defect of sheep within 12 weeks [31], and a rat femoral defect was healed completely within 10 weeks [32]. Compared with using EPCs alone, the combination of EPCs with other stem cells has been proven to promote more effective osteogenesis. In research of critical-sized bone defects in rats, the combination of MSCs and EPCs was shown to have a synergistic effect on bone healing, and the initial neovascularization by EPCs was vital for subsequent complete osteogenesis [33]. Based on the multicellular nature of bone healing, the combination of stem cells not only promotes osteogenesis but also induces angiogenesis, offering a new avenue for BTE.

### 2.3. Induced Pluripotent Stem Cells (iPSCs)

As a burgeoning seed cell in BTE, the application of iPSCs is promoted by the emergence of reprogramming measures, disease modelling, and preclinical trials [34]. Through genetic reprogramming of adult somatic cells, iPSCs have a differentiation potential similar to that of ESCs, which can be utilized for the repair of bone, cartilage, and osteochondral cartilage. To construct iPSCs, adult somatic cells can be reversely transduced with four pluripotent factors: OCT4, SOX2, NANOG, and LIN28 (Figure 3) [35]. Fully reprogrammed iPSCs can differentiate into various types of somatic cells, all of which have the same genetic information as the patient. 

For the application of iPSCs in BTE, iPSCs derived from patients will be re-directed to differentiate and culture on a scaffold providing structural and functional support, which would be later transplanted to the defect site (Figure 3). In a mouse model of limb ischemia, the therapeutic effect of iPSCs was found to be superior to BMSCs [36]. The greater potential of iPSCs may be due to the survival rate after transplantation and the tissue regeneration induced by the graft through cell differentiation and paracrine mechanisms. Moreover, on account of the clinical trials of patients with debilitating eye disease in Japan [37], iPSCs have been used as promising stem cells clinically. Following up, Tang et al. showed that iPSC-derived MSCs implanted on a calcium phosphate cement (CPC) scaffold could be used for craniofacial, dental, and orthopaedic prosthetic treatments [38]. Results suggested that iPSC-derived MSCs expressed typical surface antigens of mesenchymal cells, possessed the ability to differentiate into chondroblasts, osteoblasts and adipocytes, and had high cell viability after transplantation. Using cranial defects in nude rats, the iPSC-derived, MSC-based CPC scaffold promoted osteogenesis and accelerated scaffold resorption in vivo [38]. To further improve the vascularization of tissue-engineered bone, a tri-culture system seeding iPSC-derived MSCs with human umbilical vein endothelial cells and pericytes was examined and found to induce the vascularization of the CPC scaffolds [39]. Within 12 weeks, the favourable formation of bone and blood vessels in the skull defects of nude mice showed the viability of the co-culture system using more than one cell type to induce osteogenesis and angiogenesis [39]. The main rationale of the co-culture system is to promote the inherent capacity to form stable vascular structures and use the stem cells for osteogenesis [3]. According to Qi et al., exosomes secreted by human iPSC-derived MSCs (hiPSC-MSC-Exos) incorporated the advantages of MSCs and iPSCs without immunogenicity [40]. The hiPSC-MSC-Exos/β-TCP scaffold has been found to promote the repair of critical-sized skull defects by enhancing angiogenesis and osteogenesis in ovariectomized rat models [40]. In general, iPSCs exhibit promising osteogenesis and angiogenesis potential in BTE, but there is an urgent need to overcome the corresponding restrictions of clinical translation, such as genomic instability, immune rejection, and tumorigenesis.

## 3. Scaffolds

The scaffold of BTE provides a three-dimensional (3D) space for cell survival, tissue growth, and vascularization, and eventually, bone defects can be repaired. A successful BTE scaffold possesses osteoconductivity, osteogenicity, osseointegration, and osteoinductivity to simulate the formation of new bone [41]. Its porosity and pore size are vital factors that regulate the degradation and mechanical properties of the scaffold to optimize cell differentiation and new tissue formation [42]. Currently, natural derived biomaterials, synthetic biomaterials, and metal materials are widely used in BTE scaffolds (Table 1).

### 3.1. Naturally Derived Biomaterials

Naturally derived biomaterials are produced by living organisms, such as collagen, fibrin, silk fibrin, hyaluronic acid, polyhydroxyalkanoate and chitosan, and are eventually degraded into carbon dioxide and water (Table 1). Their good biocompatibility, vast sources, minimal adverse immunoreaction, and good plasticity lead to their participation in clinical applications of BTE [40].

#### 3.1.1. Fibrin

As a natural scaffold formed after tissue injury, fibrin can initiate hemostasis and provide an initial substrate for cell proliferation, differentiation, adhesion, and migration. It has excellent biocompatibility, controllable biodegradation, and the ability to transfer cells and biomolecules, acting as an ideal material for biomimetic bone scaffolds. Fibrin is made up of fibrinogen and thrombin, which can be extracted from the patients’ blood, allowing for an autologous scaffold [43]. In addition, the structure of fibrin substrates can be easily controlled by the concentrations of fibrinogen and thrombin. Fibrin can be injected as a liquid and then solidified in situ, which can heal bone defects of any shape. Although fibrin degrades quickly and has poor mechanical properties, various studies combined fibrin with other materials to overcome these limitations. For instance, nanomaterials have been used with fibrin to promote the bioactivity of fibrin and further mimic the nano-structural characteristics of bone [6]. Silk fibroin (SF) has been combined with a mesoporous bioactive glass (MBG) to fabricate composite MBG/SF scaffolds, which possessed superior compressive strength, good biocompatibility, and osteoinductivity [44]. Moreover, a composite poly (propylene fumarate)/fibrin scaffold has been reported to induce vascularized bone regeneration in vivo [45].

#### 3.1.2. Collagen

Collagen is considered an ideal material for BTE scaffolds because of its low antigenicity, cytocompatibility, and tissue regeneration potential. As a natural polymer, collagen can be extracted from animals, such as pig skin and rat tails. Proteolytic treatment using pepsin is the most common extraction process, which dissolves collagen crosslinks and terminal peptides, making collagen non-immunogenic. Natural collagen possesses amino acid sequences which can be used for cell bio-identification in collagen-based scaffolds [46]. However, due to its high hydrophilicity, collagen has poor mechanical properties and is prone to swelling after implantation. Hence, the modification, cross-linking, and recombination of collagen has attracted interest to make the physical, chemical, and mechanical properties of the scaffold meet the final application [47]. For instance, a calcium phosphate/collagen/hydroxyapatite composite scaffold was established, possessing a similar structure and composition to natural bone with good biocompatibility [48]. As a result, bone formation in the dorsal muscle of rabbits was detected in the sixth month after implantation. Additionally, collagen/bioceramic composite materials have been reported to enhance the mechanical properties, osteoconductivity, dimensional stability, and cell attachment [49]. Generally, collagen can improve the biocompatibility of composite scaffolds and may prove to be a suitable scaffold material in BTE, though it requires more in vivo assays to verify its feasibility [50].

#### 3.1.3. Chitosan

Chitosan is a biodegradable, biocompatible, and non-antigenic material, and thus, it has been widely researched in BTE scaffolds. Chitosan is derived from crustacean shells and can be obtained by deacetylation of chitin through alkaline hydrolysis [51]. Structurally similar to glycosaminoglycan, a significant component of bone and cartilage, chitosan is involved in the osteoblasts’ attachment, differentiation, and morphogenesis [52]. Although water insolubility, fast degradation, and poor compatibility limit chitosan’s potential to repair bone defects, its functions can be improved by combining it with other materials. According to Madhumathi et al., a composite scaffold of chitosan hydrogel and nano-hydroxyapatite significantly improved the crystallinity of the composite with good biocompatibility [53]. Moreover, porous nanocrystalline hydroxyapatite/chitosan scaffolds modified by a cold atmospheric plasma treatment have been proven to increase collagen mineralization and the infiltration of MSCs into the scaffold [54].

#### 3.1.4. Polyhydroxyalkanoates (PHA)

PHAs are materials synthesized by bacteria as energy sources in bad growth environments [55]. They show properties including biodegradability, biocompatibility, and good mechanical capacity with the potential to induce vascularization, making them great candidates for BTE [56]. Additionally, PHAs have various forms in hard tissue and soft tissue; thus, both flexible artificial blood vessels and firm bone tissues can be produced by PHAs [55]. Furthermore, PHA is easy to blend with other materials and reaches a compressive strength which is the same as true human bones (62 MPa) [57]. One of the disadvantages researchers have noted about PHA is its relatively high production cost, because the environmental condition requirement for microorganisms to produce PHA is severe [58]. However, it has been proven that some bacteria, such as Alcaligenes latus strains IAM 12664 T, can produce PHA without the limitation of any kind of nutrition, which indicates a brighter future for this material [59]. According to Lim et al., bacterial cellulose-modified polyhydroxyalkanoate (PHB/BC) scaffolds were produced to study their biocompatibility and osteogenic potential in critical-size mouse calvaria defects [56]. Results showed that the PHB/BC scaffolds could support the proliferation of 3T3-L1 preadipocytes, promote the differentiation of osteoblasts in vivo, and induce new bone formed in the 20 weeks after implantation. To summarize, PHB holds the potential to serve as the scaffold material of BTE.

### 3.2. Ceramics

The bioceramics used in BTE are inorganic compounds that can be divided into bioinert and bioactive ceramics based on their interactions with host tissues [60,61]. Bioinert ceramics, including alumina, zirconia, and silicon carbide, provide physical support without interaction with surrounding natural tissue. Conversely, bioactive ceramics, such as calcium phosphate ceramics (CPCs), hydroxyapatite (HA), and bioglass, can interact with surrounding tissues to produce a strong bone-induced response and promote the formation of new bone. CPC, which is similar in composition to the mineralized part of bone, is biodegradable and has a superior ability to induce osteogenesis, making it an excellent scaffold material for BTE. Furthermore, the surface roughness, porosity, size and solubility of CPC can affect protein adsorption, cell adhesion and osteoblast differentiation [62]. However, the low strength and high brittleness of CPC makes it difficult to be used in stress-bearing sites. Currently, the most prominent CPCs utilized in clinics include tricalcium phosphate and HA [63,64]. In addition, bioactive glass is widely used in orthopedics and dentistry [65,66]. This is because BGs, which are made of silica glass containing calcium and phosphate, possess good biocompatibility and can effectively bind biological tissues. Additionally, they could produce bioactive HA and silicon ions to promote cell differentiation and osteogenesis. Unfortunately, the absorption of BGs into surrounding tissues often takes years to complete, limiting their use as BTE stents. In summary, the bioceramics utilized in BTE possess a high compressive modulus and the capability to deliver bioactive ions, but their brittleness needs to be solved in future.

### 3.3. Metallic Materials

Compared with naturally derived biomaterials, metallic materials have excellent mechanical properties and biocompatibility (Table 1). Tantalum and titanium are the most widely studied metal materials in BTE scaffolds.

#### 3.3.1. Tantalum

Tantalum is an inert metal with anticorrosive properties, but it has a high modulus of elasticity far exceeding that of cancellous and cortical bone [67]. Therefore, tantalum scaffolds are often fabricated into a porous structure to reduce the elastic modulus and make them similar to autologous bone. At present, porous tantalum stents have been used in arthroplasty, spinal fusion surgery, foot and ankle surgery, and femoral head necrosis treatments [2]. As the trabecular structure of bone was stimulated by the porous tantalum scaffold, the outcomes confirmed its excellent biocompatibility and osteoinductivity in BTE [68]. In canine femoral shaft bone defect models, the porous tantalum scaffold integrated tightly with the host bone, and new bone formation was observed on the scaffold-host bone interface both three and six months after implantation. However, the complicated manufacturing process and slow osteogenesis contain the clinical application of tantalum to some degree.

#### 3.3.2. Titanium

Titanium and its alloys have good antiseptic properties and biocompatibility, and are widely used in BTE scaffolds, such as in total hip replacement and total knee replacement prostheses, spinal fusion cages, and bone plates [2]. However, due to their high elastic modulus, the direct use of titanium may cause bone absorption at the interface combining with the implant, resulting in the loosening of the scaffold. Therefore, titanium scaffolds are often made into a porous structure. Studies have proven that titanium has osteogenic properties. Martel-Frachet et al. found that titanium-modified scaffolds could promote the growth and proliferation of ASCs and induce the osteogenic differentiation of stem cells in the absence of GFs [69]. Nevertheless, follow-up clinical studies are needed for further verification due to the lack of long-term efficacy evaluations.

### 3.4. Synthetic Biomaterials

Synthetic biological materials utilized in BTE scaffolds mainly contain polymer organic synthetic materials, synthetic inorganic materials, and composite materials (Table 1). Synthetic biomaterials allow large-scale, precise, and designable geometry production with controllable mechanical properties and a minimal immune response [48]. The composite materials are the mainstream trend of BTE, concentrating the advantages of each material. Therefore, finding meaningful combinations of different materials is urgent.

#### 3.4.1. Polymer Organic Synthetic Materials

The polylactic acid polymer was the most widely used material in BTE [70]. There exist various artificial polymers, among which polyglycolic acid (PGA), polylactic acid (PLA), and their copolymer (PLGA) have been approved by the Food and Drug Administration [71]. PGA has been used in internal bone fixation due to its degradability, mechanical properties, and cellular viability, and the nonwoven polyglycolide scaffold functions as tissue regeneration substrates [41]. However, the slow degradation rate, hydrophobicity, and low impact toughness of PLA limit its clinical application. Thus, particle leaching and electrospinning techniques have been selected to improve the scaffolds by blending PLA with other polymers. For instance, Zhang et al. prepared PLA/octadecylamine functionalized nano-diamond composites for tissue engineering, which improved the mechanical properties of PLA [72]. The nanocomposite possessed similar properties to that of human cortical bone, because the addition of 10% wt of composites brought about more than a 200% increase in Young’s modulus and an 800% increase in hardness [72]. The modified PLGA scaffold was used to culture mouse smooth muscle cells, which showed that the cells were in good condition [73]. Moreover, the PLGA/gelatin scaffolds for the culture of mouse sciatic nerve cells resulted in good adherence and growth [74]. However, PLA, PGA, and PLGA demonstrate some shortcomings, such as poor hydrophilicity, a weak ability of cell adsorption, propensity towards aseptic inflammation, and insufficient mechanical properties.

#### 3.4.2. Synthetic Inorganic Materials

Synthetic inorganic materials utilized in BTE mainly contain calcium phosphate, bioglasses, and glass ceramics and show good biocompatibility, biodegradability, osteoconductivity, and osteoinductivity [4]. HA is considered a promising scaffold material for BTE, as it is the main component of bone salt and tooth. As an artificial synthetic, HA has excellent biosecurity, bioactivity, and affinity with low immune rejection [75]; furthermore, it has potential bone conduction and chemical stability, providing a microenvironment for seed cells to differentiate into osteoblasts. Additionally, the calcium and phosphorus of HA can participate in the body’s metabolism. While pure HA scaffolds have poor osteoinductivity, various studies selected other materials with osteoinductivity or osteogenesis capabilities to combine with HA, forming functional composite scaffolds [76]. For example, a nano-HA/chitosan/gelatin matrix improved the mechanical properties of the scaffold and promoted the proliferation and differentiation of induced gingival fibroblasts [77]. So far, the chemical synthesis methods of HA include precipitation, hydrothermal, electrochemical deposition, emulsion, and ultrasonic spray freeze drying [78]. These techniques may induce inflammatory responses and limit bone regeneration [79].

#### 3.4.3. Composite Materials

Common composite materials utilized in BTE scaffolds include calcium phosphate coating on metals, HA/poly-(d, l-lactide), HA/chitosan-gelation [4], and those containing bioceramics. They are promising candidates due to their biodegradability, osteoconductivity, compressive strength, and osteointegration properties [80]. For calcium silicate-based bioceramic components, bioactive glass (BG) with SiO_2_-CaO-P_2_O_5_ networks possesses great biocompatibility in bone and soft tissues [81]. BG releases Na^+^, Ca^2+^, and soluble silica during degradation, promoting cell proliferation and osteogenesis [82]. According to Zhang et al., the implantation of BG into a collagen scaffold could improve its angiogenic activity and stiffness in vivo [50]. Moreover, collagen/mesoporous BG nanofiber scaffolds showed more new bone formation in rats [83], suggesting that incorporating BG into collagen scaffolds could improve the osteogenic effect in vivo. 

In nature, bone adapts to the surrounding mechanical forces. The composite material adjusting to the mechanical environment has been constructed with variable modulus influenced by the force, time, and mechanical stirring frequency [84]. The piezoelectric ZnO contributes to the adaptability of the composite material, which determines the crosslinking reaction between mercaptan and olefin in the polymer composite gel to change its mechanical driving modulus. The mechano-thiol-ene polymerization promotes organo-gel remodelling, and the mechanical activation of piezoelectric ZnO results in selective polymerization, reinforcing segments within the organo-gel matrix [84]. Thus, according to the loading position, the material could adjust to its modulus and stress distribution, similar to bone remodeling behavior, and the proper combination of different materials can optimize mechanically adaptive biomaterials for the BTE scaffold.

## 4. Growth Factors

Growth factors (GFs) play an essential role in BTE, promoting cell growth and differentiation for the normal fracture healing response. GFs are commonly stored in the extracellular matrix and released after injury to affect metabolic processes through autocrine, paracrine, and endocrine signaling, binding to the receptors on target cells and then trigger intracellular signaling [85]. There are various GFs utilized during bone repair, such as bone morphogenetic proteins (BMPs), insulin-like growth factors (IGFs), transforming growth factor-β (TGF-β), and fibroblast growth factor (FGF). The efficacy of GFs depends on the dose and rate of its release in vivo, and the drug delivery systems, including vectors, cells, and gene therapy. Currently, there are various strategies for GF delivery, such as physical entrapment, hydrogel encapsulation, surface adsorption, and biomineralization (Figure 4).

### 4.1. Transforming Growth Factor β (TGF-β)

TGF-β is released from various cells, including platelets, osteoblasts, BMSCs, chondrocytes, endothelial cells, fibroblasts, and macrophages [85]. TGF-β can improve MSC proliferation, recruit the precursors of osteoblasts, induce osteoblast and chondrocyte differentiation, and produce bone matrix [85]. Additionally, TGF-β signaling can control the bone quality through perilacunar/canalicular remodelling for maintaining bone homeostasis and brittleness [86]. TGF-β has three isoforms in mammals: TGF-β1, TGF-β2, and TGF-β3 [87]. After treating BMSCs with TGF-β3, the expression of anabolism-related genes was increased, and the catabolism-related genes were decreased, suggesting that TGF-β3 can promote chondrogenesis [88]. Interestingly, the overexpression of TGF-β1 has been shown to induce chondrogenic differentiation and proliferation of human synovium-derived stem cells [89]. According to Ueda et al., TGF-β1 incorporated with a collagen sponge successfully repaired the skull defects of rabbits [90]. Moreover, Kim et al. transfected a retrovirus encoding TGF-β1 into MSCs, and found that the overexpression of TGF-β1 did not affect the cell phenotype but promoted MSC proliferation and chondrogenic ability [89].

### 4.2. Bone Morphogenetic Protein (BMP)

Approximately 20 kinds of BMPs have been identified, among which BMP-2, BMP-4, BMP-6, and BMP-7 are widely used in BTE to promote the migration of osteoprogenitors, synthesis of the matrix, and the proliferation and differentiation of seed cells [8]. BMPs are secreted from osteoprogenitors, osteoblasts, chondrocytes, and endothelial cells [85]. In 2007, BMP-2 was approved by FDA in autologous bone grafts for sinus lift surgery and atrophic jaw ridge lifts. BMP-2 has been considered to replace the traditional granular cancellous bone grafting of the anterior iliac bone to induce premaxillary cleft repairment [91]. Furthermore, the BMP-2 induced reconstruction of mandibular defects after tumour resection or osteonecrosis showed great success [92]. Nevertheless, BMP-2 treatment has certain complications—cervical fusion associated with wound infection, dysphagia, and hoarseness [93]—and excess BMP-2 might lead to osteoclast overactivation and eventually cause osteolysis and graft subsidence [94]. In clinics, to reduce the intense proteolytic activity of the implant site, high doses of BMP-2 were used in commercial scaffolds for spinal fusion, potentially leading to cancer [93]. 

BMPs have been applied with other GFs to promote osteogenesis. For example, combining TGF-β1 and BMP-7 can effectively improve chondrogenesis and superficial zone protein expression in synovial explants [95]. According to Shintani et al., TGF-β1 functions to strengthen the BMP-2-induced chondrogenesis in bovine synovial explants, preventing the downstream differentiation of hypertrophy at an early stage. Additionally, the co-administration of BMP and TGF-β can inhibit the differentiation of bone stem cells into mast cells in the early stage. Moreover, other GFs can also have a synergic effect when combined with BMPs [96]. When exposed to hypoxia during inflammation, osteoblasts release vascular endothelial growth factor (VEGF) to activate endothelial cells and promote vascular permeability [97]. Due to the coupling of angiogenesis and osteogenesis, the BMP and VEGF combination effectively forms intramembrane bone. Additionally, the co-delivery of BMP-2 and placental growth factor-2 (PGF-2) by a heparin-based nano complex can greatly promote osteogenic proliferation and differentiation [98].

### 4.3. Insulin-like Growth Factor (IGF)

IGF can promote osteoblast proliferation, bone resorption, and bone matrix synthesis. IGF is secreted from osteoblasts, chondrocytes, hepatocytes, endothelial cells, and platelets [85]. IGF-1 and IGF-2 can stimulate collagen and DNA synthesis in mouse calvariae organ cultures [99]. Additionally, IGF-1 can induce collagen and bone matrix synthesis independent of replication [100], which is administered systematically to increase bone formation but rarely works in young, fast-growing animals [101]. Compared with TGF-β, the local injection of IGF-1 greatly stimulates effects on fracture healing in a rat tibia model, and combining both factors results in higher maximum load and torsional stiffness [102]. Moreover, the synergistic interaction between platelet-derived growth factor (PDGF) and IGF-1 facilitates cutaneous wound repair better than individual PDGF, IGF-1, or FGF [103]. According to Nash et al., co-administrating IGF-1 with PDGF-BB, TGF-β, or both leads to more matrix formation in fetal rat calvariae than applying IGF-1 alone [99]. Additionally, IGF-1 was found to enhance the osteogenic ability of BMP-6 [104].

### 4.4. Fibroblast Growth Factor (FGF)

FGFs can stimulate chondrocyte maturation, bone resorption, and the proliferation and differentiation of osteoblasts. The mammalian FGF family contains 22 members produced from macrophages, monocytes, BMSCs, chondrocytes, osteoblasts, and endothelial cells. FGF-2, FGF-9, and FGF-18 are promising candidates for BTE. Specifically, FGF-2 can work in two directions for osteogenesis promotion and inhibition. High-dose FGF-2 inhibits the differentiation of osteoblasts, while low-dose FGF-2 enhances osteogenesis [104]. By activating the proliferation of osteoblasts, FGF-2 induces angiogenesis and osteogenesis in non-critical-sized bone defects. However, systemic injections were reported to cause adverse extra-skeletal effects [105]. Furthermore, according to Charles et al., a composite treatment of FGF-2 and BMP-2 demonstrated improved bone healing compared to solo BMP-2 treatment, which was shown through histological analysis of skull defect healing in mice [104]. Moreover, FGF-2 and FGF-9 are also involved in angiogenesis by controlling the expression of VEGF [106] and promoting the hypertrophy of chondrocytes [107]. According to Wallner et al., FGF-9 treatment with a collagen sponge successfully repaired unicortical defects in diabetic-model mice [108]. Interestingly, adding FGF-9 into dexamethasone-containing media has been reported to accelerate the proliferation, but not differentiation, of BMSCs [109]. FGF-18 has been illustrated to promote osteogenesis but impede chondrogenesis [110], and high dose FGF-18 could promote osteoblast differentiation in vivo, while FGF-18 treatment in vitro will inhibit mineralization [111]. According to Kang et al., a sequential delivery system was designed to release FGF-2 first and then FGF-18, which successfully repaired critical-sized bone defects of rat calvaria [112].

## 5. BTE Clinical Application and Challenges

As BTE has been extensively studied and has attracted more and more attention, the function of tissue-engineered bone could be adapted by the 3D arrangement of seed cells and the components of a proper extracellular matrix; optimized scaffold materials have been developed for different pore sizes, permeability and durability. Bone progenitor cells could be isolated and induced from various sites of the human body to possess the osteogenesis potential, and the effect and application of growth factors on osteogenesis and angiogenesis have been elucidated and well-improved in vitro and in vivo. Presently, combining stem cells, scaffolds, and growth factors, BTE has been utilized clinically in various bone defect treatments, such as traumatic calvarial defects, mandibular ridge resorption, anterior mandibular defects, and spinal stenosis (Table 2); significant amounts of promising therapeutic results have been emerging in this regard.

The biomaterial design, location and size of bone defects, health status and age of patients are vital factors in the efficiency of BTE [113]. The comprehensive consideration of these factors can improve the possibility of successful regulatory approval and commercialization of the BTE system [114]. Advancement of BTE, as alternatives to autografts, is especially called for in osteoporosis, which mainly affects ageing patients. In these patients, autologous tissues, such as ASCs, can be weakened by natural ageing or diseases, limiting their ability to regenerate new bone [34,115]. 

Osteogenesis and the integration of implants and surrounding tissues are significant issues in the therapy of bone defects [116,117]. Due to inflammation, infected bone defects may be accompanied by traumatic injury or surgical resection of tumors. In this case, construction of multi-functional BTE systems providing anti-inflammatory, antibacterial or antineoplastic drugs can help with the integration of the implants with natural tissues [118,119,120]. Versatile implants are desirable for simultaneously inhibiting biomaterial-associated infection and promoting osteointegration, especially “statically-versatile” ones with nonessential external stimulations for facilitating applications. A “statically-versatile” titanium implant, achieved by immobilizing an innovative fusion peptide (FP) containing an HHC36 antimicrobial sequence and a QK angiogenic sequence, exhibited over 96.8% antimicrobial activity against *S. aureus*, *E. coli*, *P. aeruginosa*, and *methicillin-resistant S. aureus* [36]. Simultaneously, the FP-engineered implant could enhance cellular proliferation, promote vascularization and osteogenesis. Additionally, a magnetic mesoporous calcium silicate/chitosan (MCSC) porous scaffold was reported to possess anti-tumour efficacy through the synergistic effect of doxorubicin drug release and thermal ablation. The BMP-2/Smad/Runx2 signalling pathway in the scaffolds can promote the proliferation and osteogenic differentiation of BMSCs [37]. Moreover, most reported scaffolds are limited to peripheral BTE due to the lack of timely vascularization implantation. Additionally, achieving mechanical support and mass regeneration simultaneously to reduce the dependence on existing internal and external fixation needs to be considered.

## 6. Conclusions

Based on developmental biology, morphogenesis, bioengineering, and biomechanics, BTE has developed significantly over the past few decades. The critical factors for successful BTE include stem cells, scaffolds, and GFs. 

With strong capacities to self-renew and differentiate into various kinds of offspring, competent stem cells, such as MSCs, EPCs, and iPSCs, are widely utilized as the cornerstone of BTE, boosting osteogenesis as well as angiogenesis. For the BTE scaffold, nature-derived biomaterials demonstrate excellent biocompatibility, availability, and plasticity with minimal adverse immunoreaction. Metallic materials exhibit outstanding mechanical properties and biocompatibility. Synthetic biomaterials, including polymer organic and inorganic materials, possess good versatility. Additionally, composite materials combine the advantages of each biomaterial, enabling large-scale, precise, and designable geometry production with controllable mechanical properties and minimal immune response. Among GFs, BMPs are the most essential factors in the process of bone repair. TGF-β, IGF, and FGF promote cell growth and differentiation for osteogenesis by influencing the metabolism of various cells. GFs are multitrophic, and their efficacy is related to the source, purity, dose, stem cells, culture conditions, and involvement of other GFs.

## Figures and Tables

**Figure 1 ijms-22-10233-f001:**
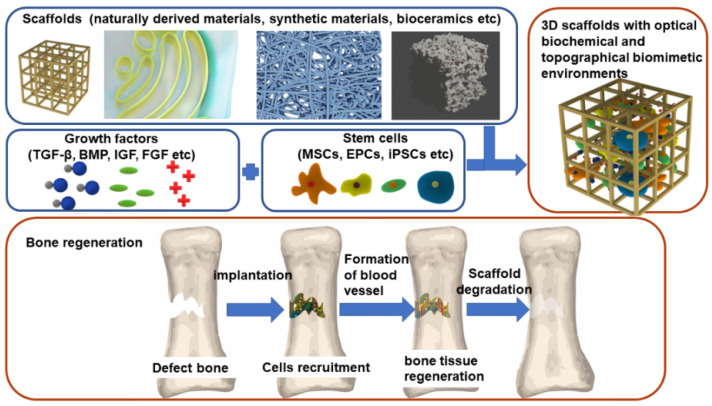
The components of BTE and the process of bone regeneration.

**Figure 2 ijms-22-10233-f002:**
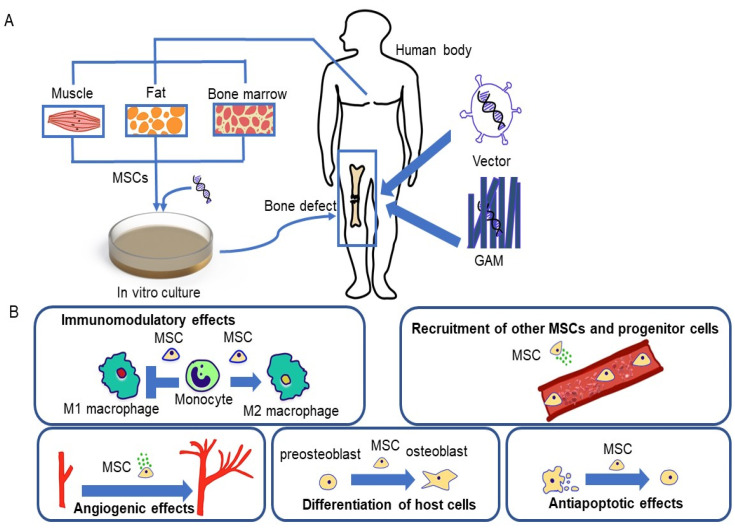
The functions, source, and genetic therapy of mesenchymal stem cells (MSCs); (**A**) Methods of gene engineering MSCs. (**B**) Functions of MSCs in bone regeneration and repair. The favorable effects include immunomodulatory effects, stimulation of angiogenesis, antiapoptotic effects on osteoblasts, recruitment of host MSCs/progenitor cells, and stimulation of their differentiation into osteoblasts.

**Figure 3 ijms-22-10233-f003:**
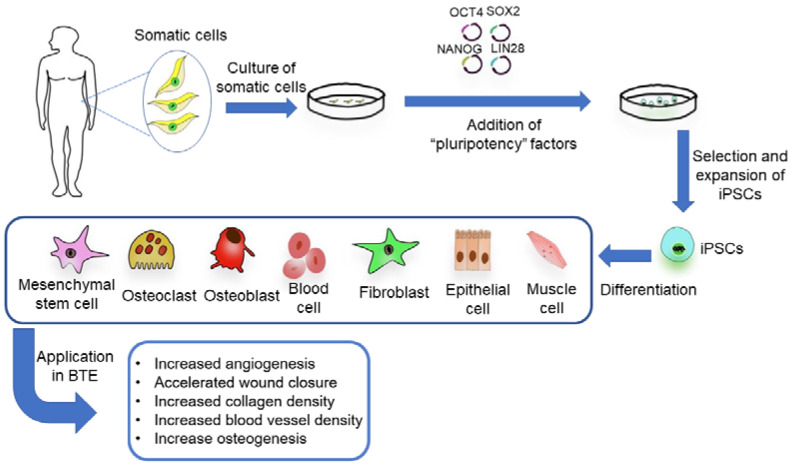
The generation and application of iPSCs in bone tissue engineering.

**Figure 4 ijms-22-10233-f004:**
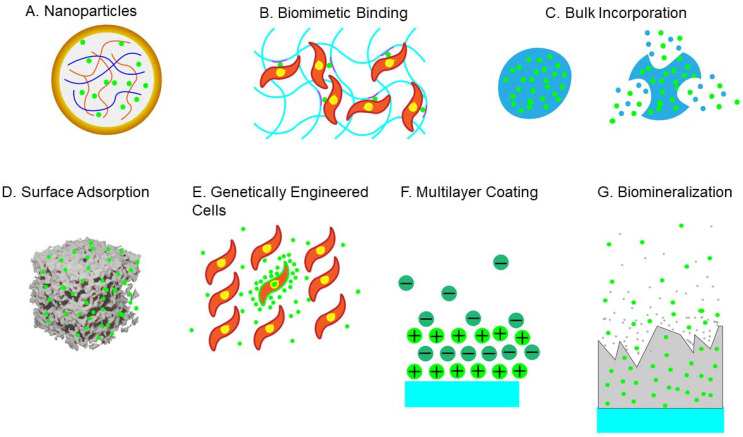
The delivery methods of growth factors (GFs) in BTE: (**A**) Physical entrapment; (**B**) Hydrogel encapsulation; (**C**) Surface adsorption; (**D**) Biomineralization; (**E**) Polyelectrolyte multilayer film coating and multiagent delivery; (**F**) Nanoparticles and macroparticles; (**G**) Cells as drug-eluting systems.

**Table 1 ijms-22-10233-t001:** The materials, examples, advantages, and disadvantages of scaffolds in BTE [4].

Bone Grafting Materials		Examples	Advantages	Disadvantages
Polymers	Natural	Protein: collagen, fibrin, silk fibrin	Biodegradability	Low mechanical strength
		Polysaccharides: hyaluronic acid, chitosan	Biocompatibility	High rates of degradation
		Bacterially synthesized poly: polyhydroxyalkanoate	Bioactivity	High batch to batch variation
			Unlimited source (some of them)	
	Synthetic	Poly-glycolic acid (PGA)	Biodegradability	Low mechanical strength
		Poly-lactic acid (PLA)	Biocompatibility	High local concentration of acidic degradation products
		Poly-(lactide-co-glycolide) (PLGA)	Versatility	
		Poly-hydroxyethylmethacrylate (poly-HEMA)		
		Poly-ε- caprolactone (PCL)		
		Poly-etylene-glycol (PEG)		
Ceramics	Calcium-phosphate	Coralline or synthetic hydroxyapatite (HA)	Biocompatibility	Brittleness
		Silicate-substituted HA	Biodegradability	Low fracture strength
		β-Tricalcium phosphate (β-TCP)	Bioactivity	Degradation rates difficult to predict
		Dicalcium phosphate dehydrate (DCPD)	Osteoconductivity	
	Bioglasses and glass-ceramics	Silicate bioactive glasses	Osteoinductivity (subject to structural and chemical properties)	
		Borate/borosilicate bioactive glasses		
	Others	Alumina ceramic (Al_2_O_3_)		
Metals		Titanium and its alloys	Excellent mechanical properties (high strength and wear resistance, ductility)	Lack of tissue adherence
		Tantalum	Biocompatibility	Corrosion
		Stainless steel		Risk of toxicity due to release of metal ions
		Magnesium and its alloys		
Composites		Calcium-phosphate coatings on metals	Combination of the above	Combination of the above
		HA/poly-(D,L-lactide)		
		HA/chitosan-gelatin		

**Table 2 ijms-22-10233-t002:** Clinical application of bone tissue engineering.

Indication	Stem Cell	Scaffold	Growth Factor	Outcome	Reference
Widespread traumatic calvarial defects	Adipose-derived stem cells	Fibrin	/	After 3 months, new bone formed with near complete calvarial continuity observed by axial and 3D-CT scans.	[18]
Severe mandibular ridge resorption	Bone marrow-derived mesenchymal stromal cells	Biphasic calcium phosphate	IGF-1, VEGF, and TGF-β	After 4 to 6 months, bone healed, as the mean volume of bone increased by 887.23 mm^3^, with little adverse events or side effects.	[121]
Large anterior mandibular defect	Adipose-derived stem cells	β-tricalcium phosphate	Recombinant human BMP-2	After 10 months, dental implants were inserted into the grafted site to allow the harvest of bone cores, and prosthodontic rehabilitation was completed based on the visualization of panoramic radiographs.	[122]
Standardized critical-size cranial defects after neurosurgery	/	Hyaluronan	BMP-2	After 3 to 6 weeks, bone was repaired with an increase in bone area of approximately 56 mm^2^, and no local or systemic side effects were observed.	[123]
Infrabony defects	Bone marrow-derived mesenchymal stromal cells	β-tricalcium phosphate	rh-PDGF-BB	6 months after surgery, the treatment resulted in a significant added benefit in terms of clinical attachment level gain (3.91 mm compared to 2.08), probing pocket depth reduction (4.50 mm compared to 3.50 mm), greater radiographic defect fill (88.33% compared to 52.77%), and improvement in linear bone growth (3.58 mm compared to 1.83 mm) in comparison to open flap debridement alone.	[124]
Spinal stenosis	Stromal vascular fraction (SVF)	β-tricalcium phosphate	/	After 6 months, the SVF/β-TCP mixture possessed higher fusion grade (3.6 compared to 2.8) and fusion rate (54.5% compared to 18.1%) than the cages filled with β-TCP. Side effects were observed in 3 out of 10 patients.	[125]
Support bone formation after sinus lift augmentation	/	β-tricalcium phosphate	Recombinant human growth and differentiation factor-5 (rhGDF-5)	The amount of new bone was between 28–31.8%. Implants failed in 4 of 47 patients (8.5%) treated with RHGDF-5/β-TCP, in agreement with the general implant failure rate of 5–15%.	[126]
Maxillary cysts	Autologous bone-derived mesenchymal stem cells	BioMax cross-linked serum scaffold	/	After 7 months, the CT density of the cyst interior increased significantly, as the mean ratio of the CT values after/before treatment was 2.52, and importantly, the density of the contralateral control area of spongy alveolar bone without treatment did not change, as the average after/before ratio was 0.99. No inflammation or other adverse effects were observed.	[127]
Intrabony defects	Autologous clinical-grade alveolar bone marrow mesenchymal stem cells	Collagen enriched with autologous fibrin/platelet lysate	/	After 12 months, the bio-complex led to significant clinical improvements for all groups with an average 3.0 mm attachment gain, 3.7 mm probing pocket depth reduction, and 0.7 mm increase in recession, without adverse healing events.	[128]

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
