# Peer review of "Current Biomaterial-Based Bone Tissue Engineering and Translational Medicine"

_ijms, 2021, doi:10.3390/ijms221910233_

Round 1
Reviewer 1 Report
The manuscript by Qi et al. “Current biomaterial-based bone tissue engineering and translational medicine” is noteworthy. Few biomaterials (grafting materials) pieces of information are missing such as polyhydroxyalkanoates. This manuscript requires moderate revision.
Comments
- The introduction, highlights the major challenges of various BTE, and the objectives of this review with prospectives.
- Please verify the numbering of Table appearance in the manuscript. Table 2 (outcome), the quantitative information may be provided to highlight significance.
- Section 3, please add one paragraph on biomaterials as polyhydroxyalkanoates with their properties and suitability as materials for BTE i.e. highly biodegradable, anti-microbial properties, and biocompatibility.
- In the text, some actual quantitative pieces of information of citations may be provided instead of just qualitative information i.e. efficiency of BTE treatment, % of repair, and period.
- Section 5, the author may highlight as “BTE clinical application and challenges”. Provide changes and perspectives of advanced BTE strategies.
Reviewer 2 Report
This is a timely and nice review on bone engineering. It has a very wide scope, which necessarily means that this review cannot dig too deeply into the specific topics, but I believe this work can still prove useful to a broad range of readers who desire to get an overview of the field. The review is well written and clear. I would probably dedicate a few more words on natural ceramic scaffold, e.g. natural apatite, because of the importance they have in the clinical practice. And I would do the same with collagen, a material that is commonly used in many clinical fields and that would probably deserve more space. The conclusion section is a bit too long to be a real conclusion: I would call that 'Dicussion' or maybe the authors can reframe it in a better way, and I would add a very brief 'Conclusion' at the end, just a passage or 2 with the take home message.
